# WDA: An Improved Wasserstein Distance-Based Transfer Learning Fault Diagnosis Method

**DOI:** 10.3390/s21134394

**Published:** 2021-06-26

**Authors:** Zhiyu Zhu, Lanzhi Wang, Gaoliang Peng, Sijue Li

**Affiliations:** 1State Key Laboratory of Robotics and System, Harbin Institute of Technology, Harbin 150001, China; zhiyuzhu2-c@my.cityu.edu.hk (Z.Z.); lisijue@hit.edu.cn (S.L.); 2The Department of Computer Science, City University of Hong Kong, Hong Kong; 3Beijing Institute of Aerospace Launch Technology, Beijing 100076, China; bluewill-926@163.com

**Keywords:** intelligent bearing fault diagnosis, Wasserstein distance, convolutional neural network, domain adaptive ability, Kuhn–Munkres algorithm

## Abstract

With the growth of computing power, deep learning methods have recently been widely used in machine fault diagnosis. In order to realize highly efficient diagnosis accuracy, people need to know the detailed health condition of collected signals from equipment. However, in the actual situation, it is costly and time-consuming to close down machines and inspect components. This seriously impedes the practical application of data-driven diagnosis. In comparison, the full-labeled machine signals from test rigs or online datasets can be achieved easily, which is helpful for the diagnosis of real equipment. Thus, we introduced an improved Wasserstein distance-based transfer learning method (WDA), which learns transferable features between labeled and unlabeled signals from different forms of equipment. In WDA, Wasserstein distance with cosine similarity is applied to narrow the gap between signals collected from different machines. Meanwhile, we use the Kuhn–Munkres algorithm to calculate the Wasserstein distance. In order to further verify the proposed method, we developed a set of case studies, including two different mechanical parts, five transfer scenarios, and eight transfer learning fault diagnosis experiments. WDA reached an average accuracy of 93.72% in bearing fault diagnosis and 84.84% in ball screw fault diagnosis, which greatly surpasses state-of-the-art transfer learning fault diagnosis methods. In addition, comprehensive analysis and feature visualization are also presented.

## 1. Introduction

With the rise of machine learning, especially deep learning, more and more data-driven algorithms have been proposed and applied successfully in different fields in the last few years [1,2,3]. Similarly, data-driven methods are increasingly suggested to deal with problems in the field of machine health monitoring [4], which has great importance in modern industry.

For example, Atoui et al. [5] presented Bayesian network for fault detection and diagnosis, Rajakarunakaran S et al. [6] proposed artificial neural networks (ANN) for the fault detection of the centrifugal pumping system, and Ivan et al. [7] suggested a novel weighted adaptive recursive fault diagnosis method based on principal component analysis (PCA) to reduce the false alarm rate in processing monitoring schemes. Recently, as deep learning is rapidly developing, artificial intelligence methods are considered to handle the fault detection and classification in rolling bearing elements, e.g., autoencoders [8] and convolutional neural networks (CNN). Li et al. [9] proposed a bearing defect diagnosis technique based on a fully connected winner-take-all autoencoder. Jafar Zarei [10] proposed a pattern recognition technique for fault diagnosis of induction motor bearings via utilizing the artificial multilayer perceptron neural networks. Olivier Janssens et al. [11] introduced feature learning means for condition monitoring based on convolutional neural networks to obtain signal features for bearing fault detection. Although many studies have been conducted, most of them are only effective under a large amount of labeled data.

Through a brief review, it is obvious that most methods are only confirmed in theory, and few are able to be applied in industry [12,13]. In a real industry situation, as machines usually work in a healthy state, it is quite a difficult task to determine whether a fault has occurred during the data collection. Moreover, even when the equipment breaking down is known, it is difficult to point out the definite fault types disassembling and inspecting the components such as bearings and ball screws in a machine, being time and labor-consuming tasks. Additionally, the real machine always works under various working conditions. Thus, the collected signals are combined with different data distributions. Those scenarios in the industrial applications will seriously impact the performance of data- based fault diagnosis.

To overcome problems of different working conditions, some researchers proposed increasing the generalization ability of the algorithms, which are named domain adaption techniques. For instance, Zhang et al. [14] proposed a deep neural network with high diagnostic accuracy in diagnosing signals with high noise and signals from different loads. In their work, the authors suggested that the high-level features of data from the different working conditions have a more similar distribution and are less affected by noise. Moreover, Zhu et al. [15] used capsule net to extract more general features from the time-frequency spectrum and achieved higher diagnosis accuracy when dealing with data from different loads. With such improvement strategies, artificial neural networks have been proven to be a potential tool to deal with industry data. However, the above methods only focus on the variation between working conditions (e.g., speed, loads) on one machine, and they cannot handle the huge variations of mechanism between different types of equipment. 

Transfer learning theory has been introduced to machine fault diagnosis in order to improve domain adaption ability among different machines. Transfer learning aims to reduce the distribution discrepancy of diverse domains, as data from the target domain have similar knowledge but different distribution compared to the source domain. For example, Lu et al. [16] presented a deep model-based domain adaptation method for the machine fault diagnosis. A gearbox dataset collected under different operation conditions was used to test the performance of the proposed method. Wen et al. [17] set up a new deep transfer learning method for fault diagnosis. The validation dataset was acquired from a bearing testbed operating under different working conditions. Xie et al. [18] proposed a transfer analysis-based gearbox fault diagnosis method. The performance of the presented method was verified by a gearbox dataset obtained under various operation conditions. Guo et al. [19] proposed deep transfer learning-based methods using maximum mean discrepancy and adversarial training techniques together to regularize the discrepancy between different domains. Sandeep et al. [20] presented a ConvNet-based transfer learning method for bearing fault diagnosis with varying speeds. Hasan et al. [21] proposed a transfer learning fault diagnosis framework using 2D acoustic spectral imaging-based pattern formation method. Zhang et al. [22] introduced hybrid-weighted adversarial learning to address the domain adaptation problem. Meanwhile, Zhang et al. [23] also utilized federated learning to facilitate the mechanical fault diagnosis. However, the above transfer learning methods took advantage of enough labeled data. Unfortunately, labeled signals from the practical industrial machine are rare and hard to collect.

As the most critical issue during the process of transfer learning, modeling and optimizing the discrepancy between different domains are the core of the proposed method. As a stable and continuous measurement, Wasserstein distance has displayed its superiority in different applications, e.g., image generation [24,25]. Thus, in this paper, we propose a new method with excellent domain adaptive ability based on Wasserstein distance (WDA) in order to deal with machine fault data from different machines. Cosine similarity and the Kuhn–Munkres algorithm are introduced to improve transfer effects. The contributions of this paper mainly lie in the following two parts:

(1) To achieve classification on unlabeled signals, we propose a transfer learning fault diagnosis method named WDA, which makes use of labeled signals from different machines to help the classification of signals. In WDA, Wasserstein distance is applied to manage the gaps between two distributions, during which we utilize cosine similarity to measure the discrepancy between feature embeddings. Moreover, Kuhn-Munkres algorithm is introduced to directly optimize the Wasserstein distance.

(2) We carried out extensive experiments to validate the effectiveness of the proposed method on various transfer scenarios. Meanwhile, to better illustrate the training process of high-dimensional feature embeddings, we also visualized the whole training process.

The structure of this paper is organized as follows. In Section 2, we introduce the basic conception of transfer learning, Wasserstein distance, and the corresponding Kuhn-Munkres algorithm. Following that, the proposed method and optimization algorithm are discussed in Section 3. Then, the experiments are carried out in Section 4 to verify the proposed method. Finally, the conclusion is drawn from the above experiments.

## 2. Related Works

In the field of machine learning, transfer learning is proposed to deal with the differences between the signals from the source domain and target domain, while Wasserstein distance is a powerful criterion of the discrepancy. However, the calculation of Wasserstein distance belongs to the general assignment problem. Yet, in most of the research work [26,27,28], there has hardly been one direct calculation of it. Thus, a brief introduction of transfer learning, Wasserstein distance, and the solution of the general assignment problem (GAP) are helpful to know about the development and the limitation of recent works.

### 2.1. Transfer Learning

Transfer learning is different from many other traditional machine learning methods, which are established under the assumption that training data and test data are drawn from the same distributions. To better illustrate transfer learning, we introduce two important conceptions: domain and task, as follows [29].

To begin with, domain D includes two key components: feature space χ and marginal distribution P(X), where $X=\left\{x_{1},\ldots ,x_{n}\right\}\in \chi$ means that X is a set containing samples from feature space χ, e.g., the signals collected from the machine in different health conditions. Then, a task consists of two components: a label space Y and an objective function G(·), corresponding to the health conditions of signals and classification algorithm. Generally speaking, the objective function could not be directly observed. However, it could be learned from training data, which consist of pairs $\left\{x_{i},y_{i}\right\}$. with the notion of source domain data $D_{s}=\left\{\left(x_{S1},y_{S1}\right),\ldots ,\left(x_{Sns},y_{Sns}\right)\right\}$ and target domain data $D_{T}=\left\{\left(x_{T1},y_{T1}\right),\ldots ,\left(x_{Tnt},y_{Tnt}\right)\right\}$. The transfer learning could be defined as the following:

Given source domain $D_{S}$ and learning task $T_{S}$, a target domain $D_{T}$ and learning task $T_{T}$ transfer learning aims to help improve the performance of the predictive function $f_{T}\left(\cdot \right)$ in $D_{T}$ through using the knowledge in $D_{S}$ and $T_{S}$, where $D_{S}\neq D_{T}$ or $T_{S}\neq T_{T}$.

In the field of fault diagnosis, source and target domains usually are different. However, the tasks are equivalent, i.e., $D_{S}\neq D_{T},{\;T}_{S}=T_{T}$. This kind of problem is also called domain adaptation, belonging to transductive transfer learning [29,30]. For the transfer learning problems, there are four different approaches to solve them: instance transfer, feature representation transfer, parameter transfer, and relational knowledge transfer. Among them, the feature representation transfer is a widely used transfer learning method in transfer fault diagnosis [18,19,31,32,33]. Moreover, there are currently two methods to bridge the gap between two distributions: feature extractor regularization, applying regularization terms on feature extractor to obtain features extracted from different domains in similar distributions, or using adversarial training methods to close two distributions.

Firstly, maximum mean discrepancy [34,35] and Wasserstein distance are widely used to measure discrepancies in domain adaptation transfer learning. They are used to regularize the output feature of the feature extractor to obtain equivalent marginal distribution. Secondly, some adversarial training methods such as DANN [36] are also proposed to narrow the gap between source and target domain. Most of them use adversarial training techniques in artificial neural networks to manage the gap of two different distributions. However, these training methods suffer problems, e.g., those methods are hard to train [37,38] and converge to a high-performance result. Thus, a high accuracy method is badly needed.

### 2.2. Wasserstein Distance

Wasserstein distance, also called earth mover’s distance, is a metric to measure the discrepancy between two distributions, and it is widely used in domain adaptation, e.g., WGAN [24] and BEGAN [39]. Wasserstein distance is generally based on a way that transforms one distribution to the other with minimal cost.

As shown in Figure 1a, different discrepancies of two domains are represented, which could also be considered as the cost of transporting distribution from one domain to the other. We define the transporting cost as:
(1)$$\ell =\frac{1}{n_{a}+n_{b}}\left(\sum_{i=1}^{n_{a}}ℱ\left(\ell_{ai}\right)+\sum_{j=1}^{n_{b}}ℱ\left(\ell_{bj}\right)\right)$$
where $n_{a}$,$\;n_{b}$ denote the numbers of samples of different fault types, and ℱ(·) represents a function measuring the difference between two samples, which usually is L2-norm or L1-norm. As shown in Figure 1b, Wasserstein distance (noted as $\ell_{1}$) is used to transport the feature from the source domain to the target domain with minimal cost. The other transport method, e.g.,$\;\ell_{2}$, shown in Figure 1b, is higher than $\ell_{1}$.

The formula of Wasserstein distance ($\ell_{w}$) is shown as:(2)$$\ell_{w}=inf_{\gamma \in \Gamma \left(\mu_{1},\mu_{2}\right)}E_{\left(x_{1},x_{2}\right)\sim \gamma }\ell \left(x_{1},x_{2}\right)$$

From the above equation, we can see that the Wasserstein distance is a low bound of the cost to transform a distance between two distributions. Berthelot et al. also proposed BEGAN to optimize the lower bound of Wasserstein distance to achieve better performance on image generation [39]. Note that all the above methods are unsupervised methods. Different from supervised or semi-supervised methods, unsupervised methods do not care about the similarity of distributions of the input and output domains. However, it would remain a huge problem, especially in the beginning training stage, if the discriminator is extremely unstable. Moreover, it is difficult to use it to regularize the feature extractor. Moreover, the discriminator could only be said to safely match the 1-Lipschitz function in the features that already are trained with the discriminator. That is, with the training process going on, the distribution of the high-level features may change, and that discriminator may not correctly calculate the distance between features from two domains. Thus, the methods based on adversarial training struggle to achieve high performance.

### 2.3. General Assignment Problem and Kuhn–Munkres Algorithm

The calculation of Wasserstein distance belongs to a general assignment problem (GAP) while the samples of two distributions are equal. Considering that there are 𝓂 samples from source domain $\left\{x_{S1},\ldots ,x_{Sm}\right\}$ and 𝓃 samples $\left\{x_{T1},\ldots ,x_{Tn}\right\}$ from target domain, without loss of generality, we assume that 𝓃≤𝓂. Any target samples $x_{Tj}$ could be assigned to the source $x_{Si}$. Each pair $\left\{x_{Si},x_{Tj}\right\}$ has a cost $𝒸\left(x_{Si},x_{Tj}\right)$ to transfer from $x_{Tj}$ to $x_{Si}$. The task is to assign 𝓃 target samples to 𝓂 source samples with the minimal cost, which is also the Wasserstein distance between two distributions. Moreover, the assignment problem could be formulated as the following optimization problem:(3)$$\min\sum_{i=1}^{𝓂}\sum_{j=1}^{𝓃}𝒸\left(x_{Si},x_{Tj}\right)\cdot T_{i,j}s.t.\;0\leq \sum_{i=1}^{m}T_{i,j}\leq 1,\;\sum_{j=1}^{𝓃}T_{i,j}=1,\;T_{i,j}\in \left\{0,1\right\}.$$

The K-M algorithm [40] could be implanted through different versions: graph [41,42] or matrix [43]. Unlike the adversarial learning-based methods, which utilize discriminator to approximate Wasserstein distance of two distributions [26,28], in this section, we introduce the K-M algorithm through graph perspective, which has been applied to the applications such as multi-objective optimization [44] and role transfer [45]. Considering a bipartite graph $G=\left(X_{Si},E,X_{Tj}\right)$, where E means the edges of pairs $\left(x_{Si},x_{Tj}\right)$ and $E\in X_{Si}\times X_{Tj}$, we introduce the following three definitions:

**Definition** **1:***Neighborhood: the neighborhood of a vertices* x*is the set*${ℐ}_{G}\left(x\right)$*with all vertices sharing edges with*x*; similarly, the neighborhood of a set*X*is*${ℐ}_{G}\left(X\right)$*, whose all vertices are sharing edges with any vertices in*X.

**Definition** **2:**
*Feasible label: it is a function*
𝓫:X→R
*, which satisfies the following condition:*



(4)$$𝓫\left(x_{Si}\right)+𝓫\left(x_{Tj}\right)\geq 𝓌\left(x_{Si},x_{Tj}\right)\;\forall x_{Si}\in X_{S}\;\forall x_{Ti}\in X_{T}$$


**Definition** **3:**
*Matched/exposed: considering a match*
 M
*, the vertex*
x
*is called matched if it is a vertex in*
M
*. Otherwise, it is exposed.*


Meanwhile, $G_{l}$ denotes the subgraph of G, which contains those edges that perfectly satisfy the feasible label, such as the following:(5)$$𝓫\left(x_{Si}\right)+𝓫\left(x_{Tj}\right)=𝓌\left(x_{Si},x_{Tj}\right)$$

Moreover, $G_{l}$ contains all the vertices of G. The K-M Algorithm 1 for solving the assignment problem is shown below.


**Algorithm 1. Kuhn–Munkres Algorithm**
  **Input:** A bipartite graph $G=\left(X_{S},E,X_{T}\right)$, corresponding edge weights $𝓌\left(x_{Si},x_{Tj}\right)$
**  Output:** the perfect matching M.  **Step 1:** Generate initial labeling ℓ and match in $G_{\ell }$
  **Step 2:** If M perfect, stop. Otherwise, pick a free vertex $x_{Si}\in X_{S}$. Set $S=x_{Si}$, T=∅.  **Step 3:** If ${ℐ}_{S}\left(X\right)=T$, update labels (forcing ${ℐ}_{S}\left(X\right)\neq T$) with following Equations (6) and (7)

(6)$$\alpha_{\ell }=\underset{s\in S,y\notin T}{\min}\left\{𝓫\left(x_{Si}\right)+𝓫\left(x_{Tj}\right)-𝓌\left(x_{Si},x_{Tj}\right)\right\}$$

(7)$$\hat{𝓫}=\{\begin{array}{l} 𝓫\left(x\right)-\alpha_{\ell },\;\;x\in S\\ 𝓫\left(x\right)+\alpha_{\ell },\;x\in T\\ 𝓫\left(x\right),\;otherwise \end{array}$$
  **Step 4:** If ${ℐ}_{S}\left(X\right)\neq T$, choose $y\in {ℐ}_{S}\left(X\right)-T$:
       If y free, 𝓊−𝓎 is augmenting path. Augment M and go to 2
       If y matched, say to 𝓏, extend alternating tree: $S=S\cup^{\;}𝓏$, $T=T\cup^{\;}y$. Go to

The K-M algorithm can efficiently address assignment problems, especially small-scale ones, e.g., transfer between two mini-batch samples. Meanwhile, Wasserstein distance as a useful divergence to measure the distance between two distributions has been widely used in the field of transfer learning. However, the performances of these methods leave much to be desired. Most of them used the approximation form of Wasserstein distance instead of calculating it directly. Actually, the calculation of Wasserstein distance is an assignment problem that could compute through the K-M algorithm. Thus, we proposed a novel method using the K-M algorithm to address the discrepancy measurement of transferring between two domains.

## 3. Proposed Method

In this section, the proposed Wasserstein distance-based domain adaptive neural network (WDA) is discussed. The architecture of the neural network and the objective of WDA are introduced.

The framework of the proposed method is shown in Figure 2. Meanwhile, the detailed architecture is shown in Figure 3. WDA is composed of two parts: CNN (feature extractor) and a fully connected layer to extract features (noted as $G_{1}\left(\theta_{f},\cdot \right)$), and a full-connected layer (classifier) noted as $G_{2}\left(\theta_{c},\cdot \right)$. The aim of CNN is to extract high-level features from input data. Before high-level features are fed into the classifier, Wasserstein distance is used to regularize the features from two different domains. Thus, the CNN could extract features from different domains with similar distributions. Finally, the classifier is used to predict the health conditions of different signals.

### 3.1. Network Architecture

The architecture of WDA is shown in Figure 4. It contains two parts: feature extractor and fully connected classifier. 

As shown in Table 1, there are essentially 12 layers in the proposed WDA. The feature extractor block contains two Conv-BN-Pooling-activation modules and a full-connected layer. In addition, the classifier contains only one full-connected layer to predict the health conditions of input data. The details of WDA are shown in Table 2. Due to the different conditions of these methods, the ranges of labels vary from condition to condition, e.g., in some datasets, they are health, inner fault, and outer fault. However, in other datasets, there are four fault types: health, inner fault, outer fault, and rolling ball fault, where $N_{o}$ means the number of classes of the models, and it should be 3 or 4. 

### 3.2. Objective of WDA

In the proposed WDA, the loss function consists of two parts: classification loss ($\ell_{c}$) on the source domain $D_{S}$ and domain adaptive loss ($\ell_{w}$) between source $D_{S}$ and target domains ${\;D}_{T}$. The classification loss aims at reducing the classification error on the source domain, and the domain adaptive loss aims to bridge the gap between the source domain and target domain. In the following section, they are introduced separately.

#### 3.2.1. Classification Loss

Classification loss of WDA is a cross-entropy loss set as Equation (8), where softmax is described in Equation (9). As shown in Algorithm 1, $G_{1}\left(\theta_{f},\cdot \right)$ represents the feature extractor and $G_{2}\left(\theta_{c},\cdot \right)$ represents the classifier. Note that classification loss is only acted upon a source domain data whose labels are known.
(8)$$\ell_{c}=\frac{1}{n}\underset{i\in n}{\sum }-\log\left(softmax\left(G_{2}\left(\theta_{c},G_{1}\left(\theta_{f},x_{s}^{i}\right)\right)\right)\right)\cdot y_{s}^{i}$$
(9)$$softmax\left(z_{i0}\right)=\frac{exp\left(z_{i0}\right)}{\sum_{i\in num\_class}exp\left(z_{i}\right)}$$

#### 3.2.2. Domain Adaptive Loss

Usually, for the semi-supervised problems, most methods want to regularize the feature extractor to obtain the features of source and target domains in exactly the same distribution. However, it is too strict for transfer learning categorical models. Actually, for the classifier, especially the linear classifier, the real concern is the pattern of the features (e.g., orientation of features). We demonstrate it through the following equation: (10)$$f\left(W,b,s_{i}\right)=Ws_{i}+b$$

As shown in Equation (10), a linear classifier, which is designed to classify output features from a feature extractor, is present to explain the mechanism, where $\;W\in R^{n,m}$ indicates the weights of classifier, $b\in R^{m}$ means the bias of the classifier, and $\;s_{i}\in R^{n}$ indicates the input feature of the classifier. If the label of the feature $s_{i}$ is $y_{i}$, Equation (11) would be established:(11)$$f{\left(W,b,s_{i}\right)}_{yi}>f{\left(W,b,s_{k}\right)}_{yk}\;y_{k}\neq y_{i}.$$

As we utilize the ReLu activation function, there is an interesting characteristic that $ReLU\left(\alpha s_{i}\right)=\alpha ReLU\left(s_{i}\right)$ (for $s_{i}\leq 0$,$\;ReLU\left(\alpha s_{i}\right)=0=\alpha ReLU\left(s_{i}\right)$). From Equation (12), we can see that if the label of the feature $s_{i}$ is $y_{i}$, the prediction of feature $\alpha \cdot s_{i}$ in the same orientation with $s_{i}$ is also $y_{i}$, that is:(12)$$f{\left(w,b,\alpha \cdot s_{i}\right)}_{yi}=\alpha \cdot f{\left(W,b,s_{i}\right)}_{yi}>\alpha \cdot f{\left(w,b,s_{k}\right)}_{yk}=f{\left(w,b,\alpha \cdot s_{k}\right)}_{yk}.$$

Equation (12) shows that the scale of features actually does not affect the classification result. Thus, the traditional ways are limited by using the L2-norm to measure the disparities between two variables. It is noted that cosine similarity calculates the orientation divergence of two vectors, which focuses more on the output pattern. Thus, for the feature extractor, we could use the cosine similarity to measure samples from different domains and change Wasserstein distance as:(13)ℓw=infγ∈Γ(μ1,μ2)E(x1,x2)~γℂ(x1,x2)
where ℂ(x1,x2) is cosine similarity from feature x1,x2, shown as the following:(14)ℂ(x1,x2)ℂ(x1,x2)=arccos(x1·x2‖x1‖·‖x2‖)1π

Once the objectives and architecture of WDA are established, the optimization of the proposed method is introduced in the following section.

### 3.3. Optimization of WDA

Following the establishment of the architecture and objective of WDA, the training algorithms are introduced in this chapter. The optimization algorithm is shown in Algorithm 2.


**Algorithm 2. Training WDA with ADAM optimization method**
$N_{c}$
**= number of categories**
Initialize: initial WDA feature extractor parameters θf and classifier parameters θgFor the number of training iterations, do:• Sample minibatch of samples (XSi,YSi)=({xS1…xSn},{yS1…ySn}), from source domain signals distribution PS(X,Y), XTj={xT1…xTn} from target domain PT(X).• Extract feature from two different domains with two shared weights feature extractors through Equation (15).
(15){χSi=G1(θf,XSi)  χTi=G1(θf,XTi)  • Calculate the n×n cost matrix A between the high-level features from source and target domains (Equation (16)).
(16)A[i,j]←ℂ(χSi,χTj) for iϵ[1,N],jϵ[1,N]• Use the K-M algorithm in Table 2 to address the assignment problem of cost matrix A.   Input: bipartite graph G=(XS,E,XT=χTi),
(17){XS={χS1,χS2,…,χSn}  XT={χT1,χT2,…,χTn}  E=A   Output: permutations S.After obtaining optimal permutations S={S1,S2…Sn}, calculate Wasserstein distance ℓd:
(18)ℓd←∑i=1nA[i,Si]• Calculate cross-entropy classification loss on the source domain.
(19)ℓc←1n∑i=1n−log(softmax(G2(G1(xSi,θf),θg))[ySi])• Calculate cross-entropy loss on the source domain.• Calculate loss.
(20)ℓ=ℓc+ℓd• Backward propagation of ℓ, getting the gradients of parameters and updating the parameters θf, θg.

**end**


Moreover, in order to verify the choice of cosine similarity, we carried out experiments that contain feature visualization and comparisons with state-of-the-art domain adaptive transfer learning methods.

## 4. Case Study and Experiment Result

In this section, experiments and analyses of the model that were carried out are shown. In order to verify the generalization of the proposed method, we separately investigated transfer scenarios on different mechanical parts, bearing and ball screws. WDA was written in python 3.6, Pytorch 0.4.1 training with Intel i3-8100 CPU, and a GTX1070 GPU.

### 4.1. CASE I: Bearing Fault Diagnosis

In this section, the proposed method was trained and tested on three different domains. There were three datasets named IMS dataset (α), self-collected bearing dataset (β), and CWRU bearing dataset (γ). We first give a brief introduction to those three datasets. Then, we present the data preprocessing procedures with implementation details and finally discuss the experimental results.

#### 4.1.1. α: IMS Bearing Dataset

The data were generated by the NSF I/UCR Center for Intelligent Maintenance Systems (IMS) [46]. These sets of data contain four bearings that were run to failure under a constant load as shown in Figure 5a,b. Every 10 min, 1 s vibration signals were collected and saved into a file that contains 20,480 points for each bearing. IMS contains four different conditions: health, inner fault, rolling elements fault, and outer fault. Radial load is 6000 lbs., and rotation speed is kept constant at 2000 RPM under all conditions.

#### 4.1.2. β: Self-Collected Bearing Dataset

The second dataset was collected by the test rig shown in Figure 6. It contains an induction motor, an accelerometer, and a rotation shaft with two bearings for support. Bearing is in the type of 6204. The dataset contains three different health conditions: health, inner fault, and outer fault as shown in Figure 7. The dataset includes artificial defects, which are shown in Figure 6. Different rotation speeds were also collected, including 900 RPM, 1020 RPM, 1140 RPM, 1260 RPM, 1380 RPM, and 1500 RPM, while the sample rate was 48 kHz.

#### 4.1.3. γ: CWRU Bearing Dataset

Data from dataset γ were collected from Case Western Reserve University [47], whose test rig is shown in Figure 8. All faults in the dataset arise in the form of EMD. The experimental setup mainly contained an induction motor, an accelerometer, testing bearings, and a loading motor.

Each bearing was tested under four different loads (1, 2, and 3 hp). In addition, damages caused by EMD lie in the outer race, inner race, or rollers of the bearings with fault diameters of 0.007, 0.014, and 0.021 inches (1 in. = 25.4 mm), respectively, which means that the number of categories under each load is 10. All of the information is listed in Table 3.

Data preprocessing and implementation details: In the proposed method, the short-time Fourier transform is applied to the raw signals to obtain a time-frequency graph. For a window sliding on the raw signals at the same stride, we obtained the signals in the window and applied Fourier transform to it. With the above steps occurring, we could change a series of time-domain signals to a graph that fuses both time and frequency features. In order to reduce the accidental noise, we applied normalization to the time-frequency graph as:(21)$$x^{*}=\frac{x-\mu }{\sigma }$$
where x is input signals, and μ and σ are the average and standard deviation of the data, respectively. Through zero-mean normalization, the effect of the noise and zero drift on the data could be removed.

In Figure 9 and Table 2, different datasets contain different signals collected in different sample rates. The sample rate greatly affects the characteristics of signals. Moreover, it is fixed for one dataset and artificially set. Thus, in the experiment, the signals were resampled to be the same (1 Kh). Meanwhile, short time Fourier transformation (STFT) was used as means of preprocessing. The kernel size of STFT was set to 128, and the stride was 5. Moreover, the size of the output time-frequency graph (TFG) was 128×63. Then it was clipped to 63×63 because TFG is symmetrical, and the first element was the dc component. Thus, the length of raw signals of a TFG was 128 + 62×5 = 438. Due to the fault types of different datasets: β, γ contained four health conditions, α contained three health conditions, and the number of samples in datasets was changed. The sample number of each health condition was 5000, e.g., in the transfer condition γ→α, the numbers of samples in train and test domains both were 20,000; however, in α→β, they were 15,000. All the training and testing signals were randomly sampled from datasets.

During the training phase, we utilized the ADAM optimizer with the first and the second momentums as 0.9 and 0.999, respectively. We trained it for 100 epochs with the batch size set to 128. The learning rate was initialized to 0.003 and exponentially decayed with a factor of 0.98 for each epoch to 0.00013. The comparison methods are listed in Table 4. The detailed information of training process of different methods is presented in Table 5. CNN denotes a simple convolutional network without any transfer learning technique. SVM represents a support vector machine [48], which is also only trained on the source domain. DDC [34] and DANN [36] are image-based transfer learning algorithms. For fair comparison, we trained them with the time-frequency graph (TFG), which is the same as the proposed method. DCTLN [17] is a transfer learning-based deep neural network for bearing fault diagnosis. We trained those methods with the same protocols and recommended hyper-parameters from the original paper for a fair comparison.

The experiment results in Figure 10 and Table 6 show the excellent performance of the proposed method. No matter the traditional machine learning method or deep learning method, it is easy to obtain semi-supervised methods that could achieve better performance than supervised methods. DCTLN as a transfer learning method designed for fault diagnosis showed its superiority over general transfer learning algorithms such as DCC and DANN. However, proposed WDA exceeded DCTLN in most conditions, especially conditions γ→β (from 80.60% to 98.96%). Although DCTLN achieved 89.70% on the γ→α, it only exceeded 1.67% to WDA. Moreover, WDA finally achieved an average accuracy of 93.72%, more than 7.79% of DCTLN, and 16.45% and 22.87% of DANN and DCC, respectively.

### 4.2. CASE II: Ball Screw Fault Diagnosis

In order to further investigate the domain adaptive ability of the proposed method, we set up a test rig for ball screw fault diagnosis, and some vibration signals were collected from the machine, which is shown in Figure 11.

In order to simulate different working conditions, we collected the vibration signals of the ball screw under different forms of end supports. As shown in Table 4, there WERE two ball screw supporting forms: fixed-floating (ζ) and fixed-none (η). In the ζ set, the ball screws were fixed in one end and supported in a floating form on the other end. In the η set, the ball screws were fixed on one end and had no support on the other end.

As seen in the Figure 12, all the transfer learning methods surpassed the traditional methods, showing that transfer learning is essential and effective to bridge the gap between different domains. Moreover, the experiment result shows that WDA had better performance than other state-of-the-art transfer learning methods. Compared to DCTLN, WDA improved about 11.04% more than DCTLN and 21.52% more than DANN. In the condition (‘ζ→η’), WDA reached an accuracy of 89.13%, greatly surpassing other methods. All this evidence shows that WDA as a transfer learning method has superiority over state-of-the-art methods.

### 4.3. Feature Visualization

To further investigate the inner mechanism of the proposed method, we applied the feature visualization to output features of CNN from both source and target domains. Different colors represent different features from different health conditions, and the shapes of feature points represent different domains.

In the feature visualization of Figure 13, the features in the whole training process of WDA are shown. Figure 13a–f indicates the visualizations of intermedia feature maps by PCA with the proposed fault diagnosis method from 0 to 100 epochs. The features in the training process gradually gathered into several lines, as in the WDA, cosine similarity was chosen to measure the differences of features from different do-mains. In addition, under the restriction of cosine similarity, features with the same characteristics (e.g., within the same category) turned to keep in the same line rather than a point, although there were some features that did not get in the same line with other features. However, this had little effect on the prediction accuracy of the proposed method. Moreover, we could see that with the training process going on, the source domain features were gradually grouped in several lines with target domain features. 

## 5. The Limitation and Future Works

The proposed method utilized the labeled source domain signals with unlabeled target domain signals for joint training. However, signals from target domains were still class-balanced, which is the limitation of the proposed method. In future works, we will continue to work on the transfer learning task to improve its practicability on the class-unbalanced signals. Moreover, we also want to popularize this method into other transfer learning background problems.

## 6. Conclusions

In order to produce a more accurate fault diagnosis in unlabeled data, we proposed a Wasserstein distance-based transfer learning fault diagnosis method called WDA. In WDA, the K-M algorithm was introduced to directly calculate the Wasserstein distance. Unlike other methods that use L2-norm measuring the Wasserstein distance, in our methods, cosine similarity was used instead. Moreover, the conception of transfer learning and Wasserstein distance were well explained. Experiments showed that: (1) WDA had better performance than state-of-the-art transfer learning fault diagnosis methods and reached average accuracies of 93.72% and 84.84% on different mechanical parts transfer learning; (2) feature visualization also demonstrated that cosine similarity is efficient to group features from different domains; and (3) the proposed methods could make use of available labeled signals to help unlabeled data classification, thus addressing the problem of the high cost of data labeling and insufficient labeled data. In the age of big data, with the cost of data labeling going up, making use of unlabeled data has become a hot research topic. Thus, transfer learning fault diagnosis requires more attention in research. 

## Figures and Tables

**Figure 1 sensors-21-04394-f001:**
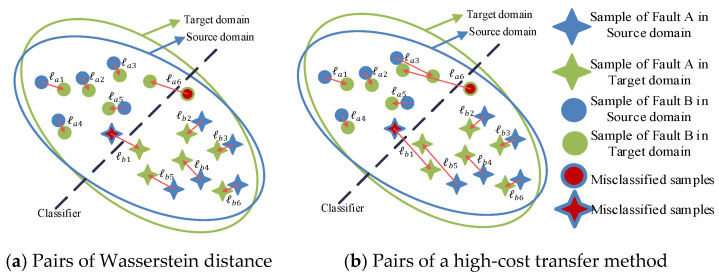
Different pairs for measuring discrepancy between two distributions.

**Figure 2 sensors-21-04394-f002:**
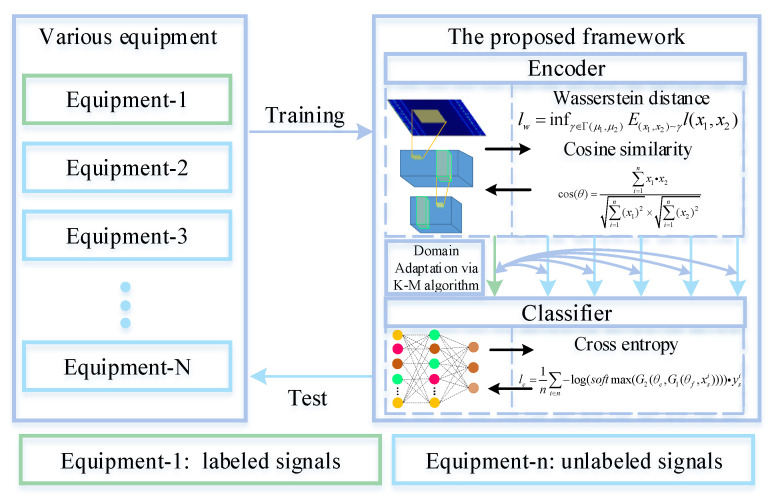
Flow chart of the proposed framework for fault diagnosis on different working equipment. During the training phase, we narrow the gaps between the distributions. At the test phase, the WDA directly predicts the health conditions of unlabeled signals.

**Figure 3 sensors-21-04394-f003:**
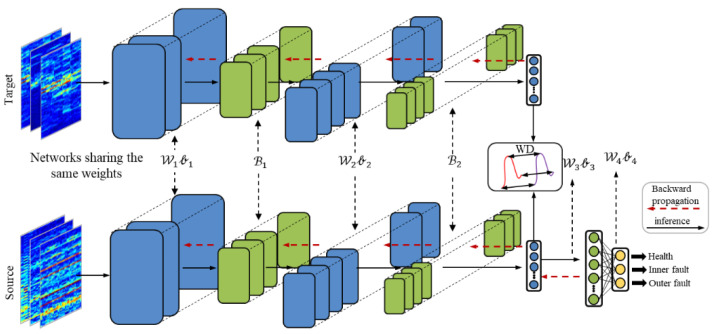
The architecture of the proposed WDA. In the testing phase, the feature maps from the target domain are directly fed into the classifier.

**Figure 4 sensors-21-04394-f004:**
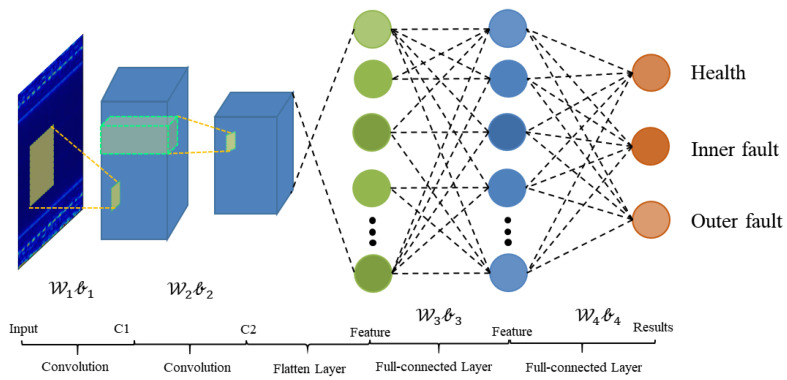
Architecture of WDA.

**Figure 5 sensors-21-04394-f005:**
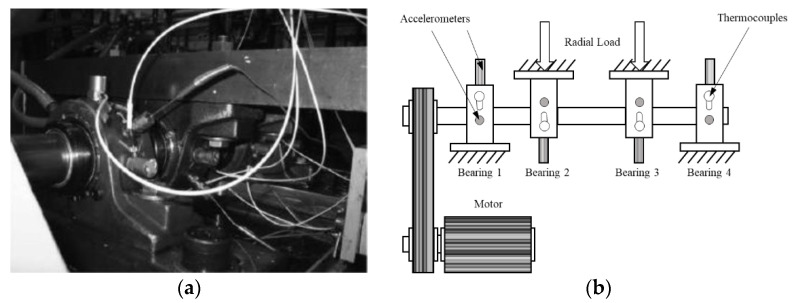
(**a**) Test rig of IMS dataset. (**b**) Illustration of IMS test rig.

**Figure 6 sensors-21-04394-f006:**
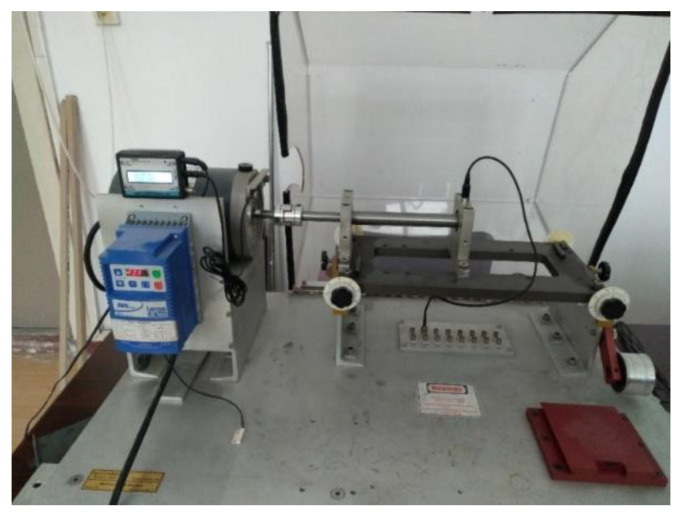
Test rig used to collect different speeds and different sample rate data.

**Figure 7 sensors-21-04394-f007:**
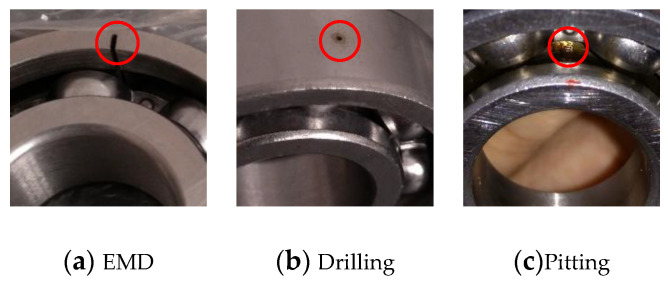
Different fault in testing bearings.

**Figure 8 sensors-21-04394-f008:**
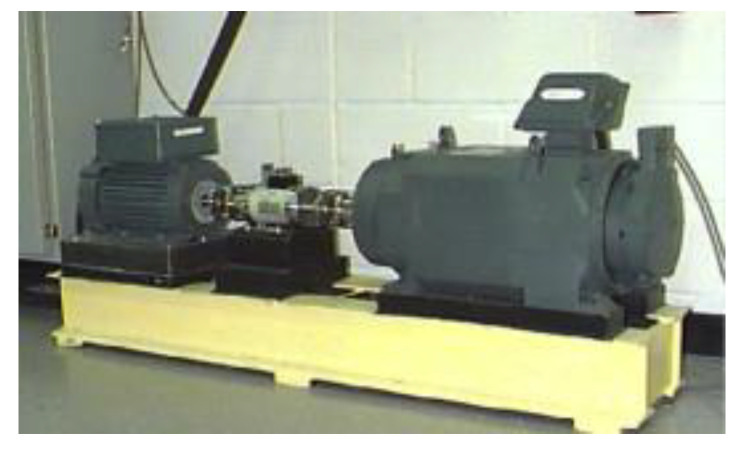
Test rig used in Case Western Reserve University Lab [41].

**Figure 9 sensors-21-04394-f009:**
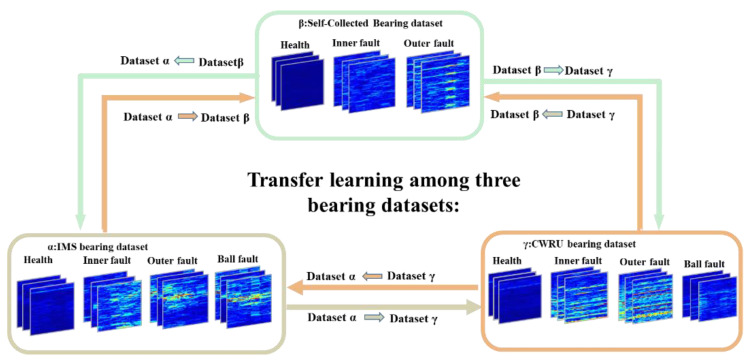
Transfer pipeline of the proposed framework. Dataset α→ dataset β denotes that we utilized the α as the source domain and the β as the target domain.

**Figure 10 sensors-21-04394-f010:**
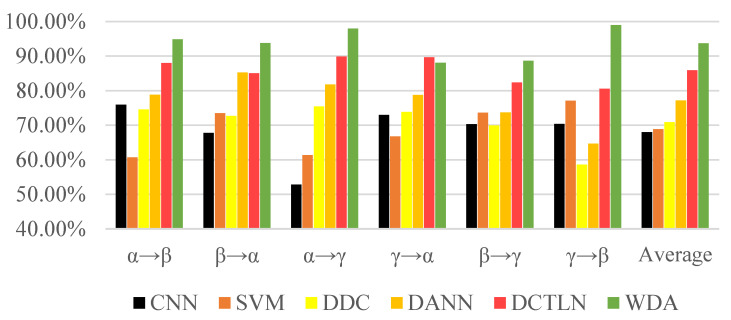
Comparison accuracies of different methods.

**Figure 11 sensors-21-04394-f011:**
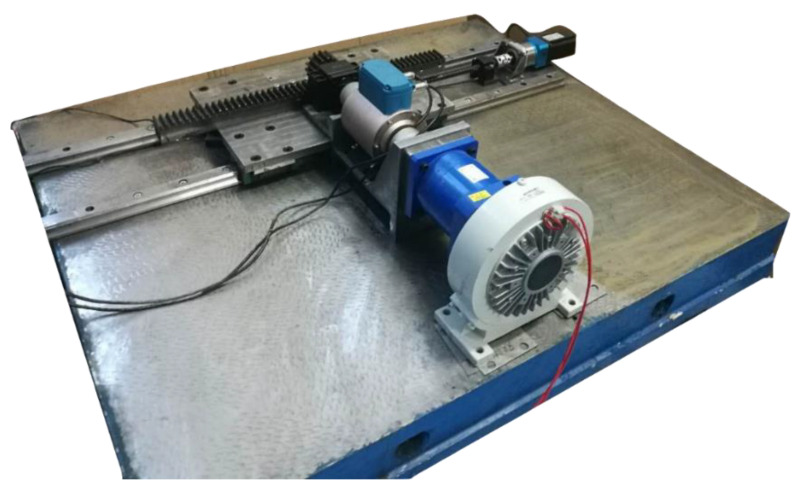
Test rig for ball screw fault diagnosis.

**Figure 12 sensors-21-04394-f012:**
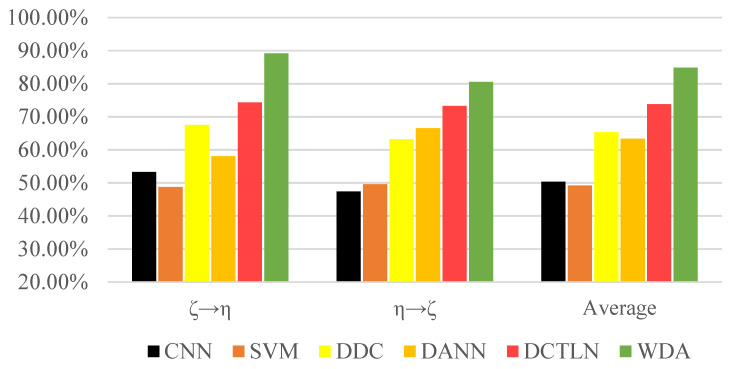
Accuracies of methods under different transfer conditions.

**Figure 13 sensors-21-04394-f013:**
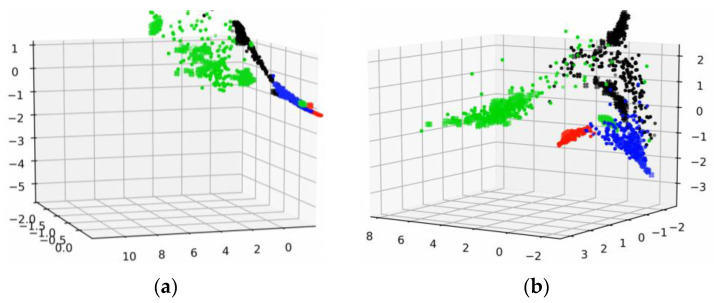
Feature visualization of the training process. (**a**) 0 epochs. (**b**) 20 epochs. (**c**) 40 epochs. (**d**) 60 epochs. (**e**) 80 epochs. (**f**) 100 epochs.

**Table 1 sensors-21-04394-t001:** Details of the classifier.

No.	Layer Name	Kernel Size/Stride/Filters	Parameters	Symbols	Output Shape
1	Convolution1	4 × 4/1/16	16(4 × 4 × 1 + 1) = 202	$W_{1}{𝒷}_{1}$/${ℬ}_{1}$	(60,60,16)
2	BatchNorm1	-	16 × 2 = 32		(60,60,16)
3	MaxPooling1	4 × 4/1/1	-		(30,30,16)
4	ReLU	-	-		(30,30,16)
5	Convolution2	3 × 3/1/64	64 × 16 × (3 × 3 + 1) = 10,240	$W_{2}{𝒷}_{2}$/${ℬ}_{2}$	(28,28,64)
6	BatchNorm2	-	-		(28,28,64)
7	MaxPooling2	2 × 2/2/1	-		(14,14,64)
8	ReLU2	-	-		(14,14,64)
9	Dense Layer1	-	(14 × 14 × 64 × 96 + 1) = 1,204,225	$W_{3}{𝒷}_{3}$	96
10	BatchNorm3	-	-		96
11	ReLU	-	-		96
12	Dense Layer2	-	(128 × 3 (4) + 1) = 387/(513)	$W_{4}{𝒷}_{4}$	$N_{o}$

**Table 2 sensors-21-04394-t002:** Description of the dataset.

Dataset	Sample Rate	Resample Rate	Loads	Speed
IMS	20 KHz	1 KHz	6000 lbs.	2000 RPM
Self-collected	25 KHz	1 KHz	0 lbs.	900–1500 RPM
CWRU	48 KHz	1 KHz	2 hp	1750 RPM

**Table 3 sensors-21-04394-t003:** The working condition of different loads.

Index	Supporting	Speed (RPM)	Loads (N.M)
ζ	Fixed—floating	1500 × Sin(5t)/400/1500	0/10/35
η	Fixed—none	1500 × Sin(5t)/400/1500	0/10/35

**Table 4 sensors-21-04394-t004:** Accuracies of methods under different transfer conditions.

Method	CNN	SVM	DDC	DANN	DCTLN	WDA
ζ→η	53.30%	48.78%	67.50%	58.10%	74.34%	**89.13%**
η→ζ	47.41%	49.60%	63.20%	66.53%	73.25%	**80.54%**
Average	50.36%	49.19%	65.35%	63.32%	73.80%	**84.84%**

**Table 5 sensors-21-04394-t005:** Description of the training process of different methods.

Name	Property	Input Type
CNN	Supervised (only source domain)	TFG
SVM	Supervised (only source domain)	TFG
DCC	Transfer learning	TFG
DANN	Transfer learning	TFG
DCTLN	Transfer learning	Time frequency signals
WDA	Transfer learning	TFG

**Table 6 sensors-21-04394-t006:** Results of different transfer result.

Method	α→β	β→α	α→γ	γ→α	β→γ	γ→β	Average
CNN	75.95%	67.78%	52.83%	73.01%	70.34%	70.35%	67.98%
SVM	60.72%	73.52%	61.35%	66.75%	73.65%	77.13%	68.85%
DDC	74.56%	72.71%	75.45%	73.87%	69.91%	58.61%	70.85%
DANN	78.80%	85.27%	81.80%	78.76%	73.72%	64.70%	77.18%
DCTLN	87.98%	85.04%	89.90%	**89.70%**	82.36%	80.60%	85.93%
WDA	**94.89%**	**93.80%**	**97.96%**	88.06%	**88.64%**	**98.96%**	**93.72%**

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
