# Peer review of "WDA: An Improved Wasserstein Distance-Based Transfer Learning Fault Diagnosis Method"

_sensors, 2021, doi:10.3390/s21134394_

Round 1

Reviewer 1 Report

see attachment

Author Response

We sincerely thank both the editor and reviewers for your time and efforts in handling our manuscript and the positive recommendation. 

To Reviewer #1

 General Criticism

Point1: Class-balanced training data.

Response: Thanks for the comment. The proposed method indeed utilizes the class-balanced signals for training, which we discussed in the limitation and future works.

Point2: The other comparison methods are not well introduced.

Response1: Thanks for the comments. We have added the introduction of the compared methods: Lines 380-390.

The comparison methods are listed in Table. 6. CNN denotes a simple convolutional network without any transfer-learning technique. SVM represents a support-vector machine [47] which is also only trained on the source domain. DDC [33] and DANN [35] are im-age-based transfer-learning algorithms. For the fair comparison, we trained them with the time-frequency graph (TFG), which is the same as the proposed method. DCTLN [19] a transfer learning-based deep neural network for bearing fault diagnosis. We train those methods with the same protocols and recommended hyper-parameters from the original paper for a fair comparison.

...

Point3: Contribution of utilizing cosine similarity and light-wight property.

Response3: Thanks for the comments. We have resummarized the contributions. The original efficiency was only used to state that there are no extra parameters during the network training, i.e., all the parameters trained can be used for testing.

Detailed Review

Point1: little or unlabeled data.

Response1: Thanks for the comments. We have reduced some discussion of the results from the CWRU dataset.

Point2:rewrite the formulation.

Response2: Thanks for the comment. We rewrite the formulation: Lines 70 -73.

Transfer learning aims to reduce the distribution discrepancy of diverse domains as data from the target domain have similar knowledge but different distribution compared to the source domain.

Point3: Contribution of efficiency.

Response3: Thanks for the comment. We have resummarized the contributions. The original efficiency was only want to state that there are no extra parameters during the network training, i.e., all the parameters trained can be used for testing.

Point4: Using of adversarial learning in the domain adaptation.

Response4: Thanks for the comment. We want to state that adversarial learning-based methods are very hard to train (please see lines 156-157). It indeed plays a critical role in the field of domain adaptation.

“…

But these training methods suffer problems, e.g., those methods are hard to train [43, 44] and converge to a high-performance result.

…”

Point5: Discussion of Wasserstein distance and K-M algorithm in related work.

Response5: Thanks for the comment. We have added some related discussions in Lines 202-206.

“…

Unlike the adversarial learning-based methods, which utilize discriminator to approximate Wasserstein distance of two distributions [25,27], in this section, we introduce the K-M algorithm through graph perspective, which has been applied to the applications such as multi-objective optimization [45] and role transfer [46].

…”

Point6: Fig. 2 is hard to read.

Response6: Thanks for the comment.  We have revised figure 2 by the comment.

Point7: Activation function.

Response7: Thanks for the comment.  We utilized ReLU activation function. We have revised Table 1.

Point8: Correctness of Eq.12.

Response8: Thanks for the comment.  We have added more discussion of Eq. 12 in Lines 290-291.

“…

As we utilize the ReLu activation function, there is an interesting characteristic that  (for ,).

…”

:Point9: The issue of Table. 3.

Response9: Thanks for the comment. We have corrected this typo.

Point10: The issue of Fig. 9.

Response10: Thanks for the comment. We have corrected this typo in figure 10 now.

Point11: Draw uniform Figure 10-11 and tables 3-6.

Response11: Thanks for the comment. We have revised Figures 10 -11 and Tables 3, and 6 to make them uniform.

Point12: Mistake: D-> E.

Response12: Thanks for the comment. We have revised this mistake.

Reviewer 2 Report

The paper is dealing with electrical machine diagnostics and implementing transfer learning to automate and improve the diagnostic procedures. There is a good introduction and state-of-art part. Followed by the theoretical part and experimental validation. In the view of research, there are no big faults or mistakes necessary to mention here.

There are numerous typos and mistakes in language and formatting of the text. Hence, the manuscript must be proof-read.

Fig. 4-2 is blurry and quality must be improved.

The references are very China-oriented. Although, the research problem is global and is actively addressed all around the globe. Maybe, it would make sense to show a variation of research organizations and researchers dealing with the problem, rather than focusing in only a certain area in the world. But this is just a suggestion, not actually a drawback as such...

Author Response

To Reviewer #1

Point1: There are numerous typos and mistakes in language and formatting of the text. Hence, the manuscript must be proofread.

Response1: The authors sincerely thank the reviewer for your time and efforts in reviewing our paper.  We have carefully check the typos and grammar mistakes in the paper.

Point2: Fig. 4-2 is blurry, and quality must be improved.

Response2: Thanks for the comment. We have redrawn the figure in the submitted manuscript.

Point3: The references are very China-oriented. Although, the research problem is global and is actively addressed all around the globe. Maybe, it would make sense to show a variation of research organizations and researchers dealing with the problem, rather than focusing on only a certain area in the world. But this is just a suggestion, not actually a drawback as such...

Response3: Thanks for pointing out the issue of reference. We have added some references to related articles.

Sandeep et al. [19] presented a ConvNet-based transfer learning method for bearing fault diagnosis with varying speed. Hasan et al. [20] proposed a transfer learning fault diagnosis framework using 2D acoustic spectral imaging based pattern formation method.

  1. Udmale, S.S.; Singh, S.K.; Singh, R.; Sangaiah, A.K., Multi-Fault Bearing Classification Using Sensors and ConvNet-Based Transfer Learning Approach. Ieee Sens J 2020, 20, (3), 1433-1444.
  2. Hasan, M.J.; Islam, M.M.M.; Kim, J., Acoustic spectral imaging and transfer learning for reliable bearing fault diagnosis under variable speed conditions. Measurement 2019, 138, 620-631.

Reviewer 3 Report

This paper proposes an Improved Wasserstein Distance Based Transfer Learning Fault Diagnosis Method. In general, this paper is well presented. The idea is interesting and experiments validate it. The following issues can be considered before publication. 1. The idea behind the introducing of Wasserstein distance on the fault diagnosis problem can be more discussed. That is a new perspective in using new methods on the traditional problems. 2. A flow chart of the proposed method can be added for better illustration. 3. Some recent related works on fault diagnosis can be reviewed, such as "Universal Domain Adaptation in Fault Diagnostics with Hybrid Weighted Deep Adversarial Learning", "Federated learning for machinery fault diagnosis with dynamic validation and self-supervision" etc. 4. Fig 7 can be properly referenced. 5. Discussion of the results of the CWRU can be reduced, since that is really easy dataset.

Author Response

To Reviewer #2

Point 1: 1. The idea behind the introducing of Wasserstein distance on the fault diagnosis problem can be more discussed. That is a new perspective in using new methods on the traditional problems.

Response1: The authors sincerely thank the reviewer for your time and efforts in reviewing our paper.  The authors have added some discussion for introducing the motivation of introducing Wasserstein distance into the proposed method in Lines 90-93.

Point 2: A flow chart of the proposed method can be added for better illustration.

Response2: Thanks for the comments. We have added Fig. 2 to illustrate the proposed method better.

Point 3: Some recent related works on fault diagnosis can be reviewed, such as "Universal Domain Adaptation in Fault Diagnostics with Hybrid Weighted Deep Adversarial Learning", "Federated learning for machinery fault diagnosis with dynamic validation and self-supervision" etc.

Response3: Thanks for the comments. We have added the corresponding references.

Zhang et al. [21] introduced hybrid weighted adversarial learning to address the domain adaptation problem. Meanwhile, Zhang et al. [22] also utilized federated learning to facili-tate the mechanical fault diagnosis.

  1. Zhang, W., et al., Universal Domain Adaptation in Fault Diagnostics with Hybrid Weighted Deep Adversarial Learning. IEEE Transactions on Industrial Informatics, 2021.
  2. Zhang, W., et al., Federated learning for machinery fault diagnosis with dynamic validation and self-supervision. Knowledge-Based Systems, 2021. 213: p. 106679.

Point 4: Fig 7 can be properly referenced.

Response4: Thanks for the comments. We have referred the Fig.8 (Fig.7 in previous version) in the manuscript.

“….Data from Dataset γ is collected from Case Western Reserve University [22], whose test rig shows in Fig.8…..“

Point 5: Discussion of the results of the CWRU can be reduced, since that is really easy dataset.

Response5: Thanks for the comments. We have reduced some discussion of the results from the CWRU dataset.

Round 2

Reviewer 3 Report

The comments are well addressed. It can be published.

Author Response

Thanks a lot. Your comments are greatly helpful for us to improve the paper.